# ADVERSARIAL ATTACKS AS NEAR-ZERO EIGENVALUES IN THE EMPIRICAL KERNEL OF NEURAL NETWORKS

## ABSTRACT

Adversarial examples —imperceptibly modified data inputs designed to mislead machine learning models— have raised concerns about the robustness of modern neural architectures in safety-critical applications. In this paper, we propose a unified mathematical framework for understanding adversarial examples in neural networks, corroborating Ian Goodfellow's original conjecture that such examples are exceedingly rare, despite their presence in the proximity of nearly every test case. By exploiting results from Kernel Theory, we characterise adversarial examples as those producing near-zero Mercer's eigenvalues in the empirical kernel associated to a trained neural network. Consequently, the generation of adversarial attacks, using any known technique, can be conceptualised as a progression towards the eigenvalue space's zero point within the empirical kernel. We rigorously prove this characterisation for trained fully-connected neural networks under mild assumptions on the nonlinear activation function, thus providing a mathematical explanation for the apparent contradiction of neural networks excelling at generalisation while remaining vulnerable to adversarial attacks. In practical experiments conducted on the MNIST dataset, we have verified that adversarial examples generated through the widely-used Deep Fool algorithm do, indeed, lead to a shift in the distribution of Mercer's eigenvalues toward zero. These results are in strong agreement with predictions of our theoretical framework.

## 1 INTRODUCTION

Adversarial examples are specially crafted input data points that are designed to cause a model to output an incorrect prediction. These examples are created by making small, imperceptible perturbations to the input data (e.g., images, text or audio) which are typically indistinguishable to humans but can have a significant impact on the model's output Szegedy et al. (2014); Goodfellow et al. (2015). The existence of adversarial examples has raised questions about the robustness and reliability of deep learning models when used in safety-critical applications Ruan et al. (2021). Indeed, empirical evidence shows that neural networks are particularly sensitive to adversarial examples; for example, in the context of image analysis, it has been shown that for any trained neural network and input image, it is always possible to find an imperceptible change to the image that yields a misclassification result Moosavi-Dezfooli et al. (2017)

Adversarial attacks can occur in many application domains, and these vulnerabilities can be exploited by malicious attackers Ren et al. (2020). For example, in image classification, a self-driving car's neural object detection system might misclassify a stop sign as a yield sign if an adversarial sticker is placed on it Akhtar et al. (2021). In Natural Language Processing, attackers can make subtle changes to text, such as adding or removing words or modifying individual characters, to deceive sentiment analysis or spam detection models Zhang et al. (2020b) and, in cybersecurity, attackers can modify malware to evade detection by antivirus or intrusion detection systems Rosenberg et al. (2021). As a result, researchers and practitioners are actively developing techniques to defend against adversarial attacks , such as adversarial training, input preprocessing, and robust model architectures Zhang et al. (2020a); Fowl et al. (2021); Carlini et al. (2019); Han et al. (2023).

The foundational paper introducing adversarial examples Szegedy et al. (2014) characterised them as "intriguing properties" of neural networks and raised a compelling question: how can neural networks exhibit strong generalisation performance on test examples drawn from a data distribution, while simultaneously being susceptible to adversarial examples? An interesting hypothesis, which we recapitulate next, was already outlined in the paper's concluding remarks.

"*A possible explanation is that the set of adversarial negatives is of extremely low probability, and thus is never (or rarely) observed in the test set, yet it is dense (much like the rational numbers), and so it is found near virtually every test case*."

Despite the substantial attention dedicated by researchers to explaining the prevalence of adversarial examples, as discussed in Section 6, and despite progress in identifying new attack variants and developing defensive approaches, robustness measures Yu et al. (2019); Carlini et al. (2019); Wang et al. (2023), and theoretical guarantees based on such measures Bhagoji et al. (2019); Shafahi et al. (2019), the fundamental nature of adversarial examples continues to elude complete understanding Guo et al. (2018); Ilyas et al. (2019); Madry et al. (2018); Qin et al. (2019).

In this paper, we take an additional stride in this direction and provide novel theoretical results that further substantiate the original hypothesis by Szegedy et al. (2014). Our research relies on the formulation of fully-connected neural networks (FCNs) as specific instances of parameter-dependent kernel machines, which has enabled researchers to leverage well-established results in the field of kernel theory Györfi et al. (2002); Saitoh & Sawano (2016); Berthier et al. (2020) to investigate the generalisation properties of neural networks Canatar et al. (2021); Simon et al. (2022). Specifically, a trained FCN can be equivalently viewed as realising a non-linear transformation of the input data, which is associated to an *empirical kernel*, followed by a linear transformation characterised by the *readout weights* of the network.

*Mercer's decomposition theorem* Mercer (1909); Minh et al. (2006) is a fundamental result in kernel theory which plays a crucial role in various ML algorithms and provides the mathematical foundation for understanding the properties of kernel functions. Mercer's decomposition theorem states that, for a given kernel function applicable to pairs of data points sampled from a given probability distribution, the kernel's application can be represented as an infinite sum of products. These products consist of the application of *Mercer's eigenfunctions* to the input data points, mapping them to real values, weighted by *Mercer's eigenvalues*—positive scalar values determining the relative contribution of each term in the Mercer decomposition. Importantly, such eigenvalues and eigenfunctions are specific to the kernel function and input data distribution, but remain independent from the specific data points being evaluated.

We adopt the perspective that the introduction of an adversarial example can be regarded as a modification of the empirical data distribution derived from the training examples, expanded to incorporate the adversarial example. According to Mercer's theorem, even when maintaining the same empirical kernel associated to a trained FCN, this process results in a fresh set of Mercer eigenvalues and eigenfunctions. This viewpoint enables us to characterise adversarial examples as those producing near-zero Mercer eigenvalues in the updated decomposition of the the FCN's empirical kernel. This characterisation allows us to show that adversarial examples have measure zero in the limit where they become infinitesimally close to any test example sampled from the original data distribution.

We interpret our results as providing a rigorous mathematical explanation for what appears to be a paradoxical empirical observation: while neural networks demonstrate strong generalisation to novel new test examples, they are also susceptible to adversarial attacks. Indeed, it follows from our results that adversarial examples are exceedingly unlikely to occur in the test set, thereby providing compelling support for the original hypothesis proposed by Szegedy et al. (2014).

To validate our theory, we conducted experiments on a FCN trained on a subset of the MNIST dataset LeCun (2012), achieving zero training error and perfect generalisation performance on a subset of the test set. We then used the well-known DeepFool algorithm Moosavi-Dezfooli et al. (2016) to generate a collection of adversarial examples, and resorted to the numerical method established by Baker (1977); Rasmussen & Williams (2006) to compute Mercer's decomposition on the empirical kernel associated with both the original empirical data distribution and our updated empirical data distributions, which incorporate the adversarial examples. We have plotted and analysed the relevant distributions of eigenvalues, computed those with minimal value and estimated the integral of

relevant quantities near zero. Our numerical experiments align with our theory, demonstrating that adversarial examples induce a shift in the distribution of Mercer eigenvalues towards zero.

## 2 PRELIMINARIES

**Matrices and vectors.** We use standard notation for real-valued vectors $\mathbf{v} \in \mathbb{R}^n$, matrices $\mathbf{A} \in \mathbb{R}^{m \times n}$, and their transposes $\mathbf{v}^T$ and $\mathbf{A}^T$. As usual, $A_{i,j}$ denotes the $(i,j)$-element of $\mathbf{A}$ and $v_i$ denotes the $i$-th element of $\mathbf{v}$; furthermore, we use $\mathbf{a}_i$ to denote the vector in the $i$-th row of $\mathbf{A}$. The $n$-dimensional zero vector is denoted as $\mathbf{0}_n$. For $\mathbf{A} \in \mathbb{R}^{m \times n}$ and $\mathbf{v} \in \mathbb{R}^n$, we denote with $\mathbf{A} : \mathbf{v}$ the matrix obtained by extending $\mathbf{A}$ with $\mathbf{v}$ as an additional row. The Moore-Penrose pseudo-inverse of a matrix $\mathbf{A}$ is denoted as $\mathbf{A}^\dagger$ Moore (1920) and the trace of $\mathbf{A}$ is denoted as $\mathrm{Tr}(\mathbf{A})$.

**Fully-connected neural networks.** A fully-connected neural network (FCN) with $L$ layers is a tuple $\mathcal{N} = \langle \{\mathbf{W}^\ell\}_{1 \leq \ell \leq L}, \{\mathbf{b}^\ell\}_{1 \leq \ell \leq L}, \{\sigma^\ell\}_{1 \leq \ell \leq L} \rangle$. Each layer $\ell \in \{1, ..., L\}$ consists of a *weight matrix* $\mathbf{W}^\ell \in \mathbb{R}^{N_\ell \times N_{\ell-1}}$, a *bias vector* $\mathbf{b}^\ell \in \mathbb{R}^{N_\ell}$ and an *activation function* $\sigma^\ell : \mathbb{R} \mapsto \mathbb{R}$ where $N_\ell$ is the layer's *width*. On input $\mathbf{x} \in \mathbb{R}^{N_0}$, network $\mathcal{N}$ sets $\mathbf{x}^0 := \mathbf{x}$ and then subsequently computes, for each layer $1 \leq \ell \leq L$, a sequence of *pre-activations* $\mathbf{h}^\ell = \mathbf{W}^\ell \mathbf{x}^{\ell-1} + \mathbf{b}^\ell$ and *post-activations* $\mathbf{x}^\ell = \sigma^\ell(\mathbf{h}^\ell)$. The network's output $\mathcal{N}(\mathbf{x})$ on input $\mathbf{x}$ is then given by $\mathbf{x}^L$. We assume that all layers except the last one have the same width $N$. We also assume that nonlinear activation functions $\sigma^\ell$ are Lipschitz continuous, a property that is satisfied by all commonly-used activation functions. For the last layer, we assume width $N_L = 1$ (ensuring a real-valued output), $b^L = 0$ and $\sigma^L = \mathrm{Id}_\mathbb{R}$ (thus ensuring linearity). In this setting, the weights $\mathbf{W}^L$ are referred to as the *readout weights*.

**Kernels.** A kernel on $\mathbb{R}^{N_0}$ is a positive semi-definite symmetric function $K : \mathbb{R}^{N_0} \times \mathbb{R}^{N_0} \mapsto \mathbb{R}$. By Mercer's theorem Minh et al. (2006), given a distribution $p$ on $\mathbb{R}^{N_0}$, there exist unique countable collections of *Mercer's eigenvalues* $\lambda_i^{K,p}$ and *Mercer's eigenfunctions* $\varphi_i^{K,p}$, for $i \in \mathbb{N}$, such that

- $K(\mathbf{x}, \mathbf{x}') = \sum_i^\infty \lambda_i^{K,p} \varphi_i^{K,p}(\mathbf{x}) \varphi_i^{K,p}(\mathbf{x}')$ for all $\mathbf{x}, \mathbf{x}' \in \mathbb{R}^{N_0}$; and

- for $i, j \in \mathbb{N}$, we have $\mathbb{E}_{\mathbf{x} \sim p(\mathbf{x})} \left( \varphi_i^{K,p}(\mathbf{x}) \varphi_j^{K,p}(\mathbf{x}) \right) = \delta_{i,j}$, with $\delta_{i,j}$ the Kronecker Delta.

The first condition represents the application of the kernel function to data points $\mathbf{x}$ and $\mathbf{x}'$ as an infinite sum of products, where the $i$-th product consists of the application of the $i$-th Mercer eigenfunction to $\mathbf{x}$ and $\mathbf{x}'$, mapping these data points onto real values, weighted by the $i$-th Mercer eigenvalue. In turn, the second condition requires orthonormality of the Mercer eigenfunctions with respect to the data distribution. We define the *density* $\rho^{K,p}(\lambda)$ of Mercer's eigenvalues as the measure $\lim_{M \to \infty} \frac{1}{M} \sum_{i=1}^M \delta_{\lambda_i^{K,p}}(\lambda)$, where $\delta_{\lambda_i^{K,p}}$ is the Dirac measure.

**Empirical feature maps.** Consider a FCN $\mathcal{N}$ with $L$ layers. The mapping from an input $\mathbf{x}$ to the activation $\mathbf{x}^{L-1}$ is a nonlinear transformation $\phi_\mathcal{N} : \mathbb{R}^{N_0} \mapsto \mathbb{R}^N$ called the *empirical feature map*, which is associated to an *empirical kernel* $K_\mathcal{N} : (\mathbf{x}, \mathbf{x}') \mapsto \langle \phi_\mathcal{N}(\mathbf{x}), \phi_\mathcal{N}(\mathbf{x}') \rangle$ expressed as the inner product between the corresponding feature map evaluations. By definition, $\mathcal{N}(\mathbf{x}) = (\mathbf{W}^L)^T \phi_\mathcal{N}(\mathbf{x})$ for any input $\mathbf{x}$ to the FCN.

Consider a training set $(\mathbf{X}, \mathbf{y})$ with $P$ examples sampled i.i.d. from an unknown distribution $p$. The evaluation $\phi_\mathcal{N}(\mathbf{X}) = (\phi_\mathcal{N}(\mathbf{x}_1), ..., \phi_\mathcal{N}(\mathbf{x}_P))^T \in \mathbb{R}^{N \times P}$ of the empirical feature map on the training set induces an *empirical feature covariance matrix* $\mathbf{K}_\mathcal{N}(\mathbf{X}, \mathbf{X}) \in \mathbb{R}^{P \times P}$, where element $(i,j)$ for $1 \leq i \leq j \leq N$ is given by $K_\mathcal{N}(\mathbf{x}_i, \mathbf{x}_j)$. Finally, the training data $\mathbf{X}$ also induces an *empirical probability distribution* $p_\mathbf{X}$ defined as $p_\mathbf{X}(\mathbf{x}) = \frac{1}{P} \left( \sum_{i=1}^P \delta_{\mathbf{x}_i}(\mathbf{x}) \right)$ with $\delta_{\mathbf{x}_i}$ the Dirac measure.

**Adversarial examples.** Consider a training set $(\mathbf{X}, \mathbf{y})$ sampled from an unknown distribution $p$ and let $\mathcal{N}$ be a FCN trained on $(\mathbf{X}, \mathbf{y})$ to zero error. Note that this assumption is not overly restrictive, as training to interpolation is common practice in modern deep learning Ishida et al. (2020); Belkin (2021); Mallinar et al. (2022). Let $f$ be a real-valued function such that $f(z) \to \infty$ as $z \to 0$, and let $\epsilon > 0$. A vector $\mathbf{x}'$ is an $(\epsilon, f)$-adversarial example for $\mathcal{N}$ and $p$ if there exists an example $(\mathbf{x}^*, y^*) \sim p$ such that $||\mathbf{x}^* - \mathbf{x}'|| \leq \epsilon$ and $(y^* - \mathcal{N}(\mathbf{x}^*))^2 < \infty$, but $(y^* - \mathcal{N}(\mathbf{x}'))^2 \geq f(\epsilon)$.

Function $f$ is introduced to adjust the definition of adversarial example for classification in the literature to the regression setting, where there is no a-priori notion of what it means for the adversarial example to change the prediction (in classification, changing the prediction means predicting a different class). Function $f$ allows practitioners to quantify when a modification of the output is significant for the regression task at hand. To simplify the presentation, we fix an arbitrary such $f$ and speak from now onwards of $\epsilon$-adversarial examples. The first condition in the definition requires that the adversarial example $\mathbf{x}'$ is close in norm to some test example for which $\mathcal{N}$ generalises to bounded error; the second requirement ensures that the corresponding error diverges w.r.t. $\mathbf{x}'$.

# 3 ADVERSARIAL ATTACKS SHIFT MERCER'S SPECTRUM TOWARDS NEAR-ZERO EIGENVALUES

In this section, we show that adversarial examples can be characterised as those that shift the Mercer's spectrum of the empirical kernel corresponding to a trained neural network to yield near-zero eigenvalues with sufficient probability.

To gain insight into this result, consider a FCN $\mathcal{N}$, which has been successfully trained (technically, to zero error) on a dataset $(\mathbf{X}, \mathbf{y})$ drawn from an unknown data distribution $p$. In this scenario, Mercer's theorem reveals the existence of a unique collection of eigenvalues and eigenfunctions associated to the kernel $K_{\mathcal{N}}$ and the known empirical distribution $p_{\mathbf{X}}$ derived from the training data.

Now, suppose that we sample some new example $(\mathbf{x}^*, y^*)$ according to the true data distribution $p$ for which the network exhibits reasonable generalisation performance (technically, it suffices that the error remains bounded), and consider any data point $\mathbf{x}'$ in the vicinity of $\mathbf{x}^*$. After augmenting the training set with the new example $\mathbf{x}'$, we reconsider the Mercer spectrum for $K_{\mathcal{N}}$ and the updated empirical distribution. In this context, we can characterise $\mathbf{x}'$ as adversarial if and only if the eigenvalues in the updated Mercer spectrum exhibit sufficient density near zero. Consequently, the generation of adversarial attacks, using any known technique, can be conceptualised as a progression towards the eigenvalue space's zero point within the empirical kernel.

The main insight behind the proof is the observation that the error $(y^* - \mathcal{N}(\mathbf{x}'))^2$, which quantifies the discrepancy between data point $\mathbf{x}'$ and example $(\mathbf{x}^*, y^*)$ can be written as a sum of terms involving Mercer eigenvalues and eigenfunctions in the updated spectrum. In this decomposition, the value of terms depending on the eigenfunctions can be bounded as $\epsilon \to 0$, and hence the only way to obtain a divergence in the generalisation error characteristic of adversarial behaviour is for the terms involving the eigenvalues to diverge to infinity.

**Theorem 1.** *Let $\mathcal{N}$ be a FCN trained on $(\mathbf{X}, \mathbf{y}) \sim p$ to zero error. In the limit $\epsilon \to 0$, each data point $\mathbf{x}' \in \mathbb{R}^{N_0}$ such that $||\mathbf{x}^* - \mathbf{x}'|| \leq \epsilon$ for some example $(\mathbf{x}^*, y^*) \sim p$ satisfying $(y^* - \mathcal{N}(\mathbf{x}^*))^2 < \infty$ is an $\epsilon$-adversarial example for $\mathcal{N}$ and $p$ if and only if the function $\lambda \mapsto \frac{1}{\lambda^2}\rho^{K,p}_{\mathbf{X}'}(\lambda)$ is not integrable near zero [1], with $\mathbf{X}' = \mathbf{X} : \mathbf{x}'$.*

*Proof.* Following the overall proof idea sketched above, our first step will be to derive the relevant expression for the squared error, $(y^* - \mathcal{N}(\mathbf{x}'))^2$, valid in the limit $\epsilon \to 0$.

The readout weights $\mathbf{W}^L$ of $\mathcal{N}$ can be obtained as the solution of the linear system $[\phi_{\mathcal{N}}(\mathbf{X})]^T\mathbf{W}^L = \mathbf{y}$, which is given by the following expression involving the empirical feature map $\phi_{\mathcal{N}}$ and covariance matrix $\mathbf{K}_{\mathcal{N}}$ of $\mathcal{N}$ on the training set $(\mathbf{X}, \mathbf{y})$:

$$\mathbf{W}^L = \phi_{\mathcal{N}}(\mathbf{X})[\mathbf{K}_{\mathcal{N}}(\mathbf{X}, \mathbf{X})]^{\dagger}\mathbf{y} + \mathbf{w}. \tag{1}$$

Here, $\mathbf{w}$ is an element of the null-space of $\phi_{\mathcal{N}}(\mathbf{X})$, i.e., verifying $[\phi_{\mathcal{N}}(\mathbf{X})]^T\mathbf{w} = \mathbf{0}_P$. Thus, the evaluation $\mathcal{N}(\mathbf{x}')$ of $\mathcal{N}$ on data point $\mathbf{x}'$ is given by the following expression:

$$\mathcal{N}(\mathbf{x}') = [\phi_{\mathcal{N}}(\mathbf{x}')]^T \left( \phi_{\mathcal{N}}(\mathbf{X})[\mathbf{K}_{\mathcal{N}}(\mathbf{X}, \mathbf{X})]^{\dagger}\mathbf{y} + \mathbf{w} \right) = \tilde{\mathcal{N}}(\mathbf{x}') + \mathcal{N}_0(\mathbf{x}'). \tag{2}$$

where $\tilde{\mathcal{N}}(\mathbf{x}') := [\phi_{\mathcal{N}}(\mathbf{x}')]^T \phi_{\mathcal{N}}(\mathbf{X})[\mathbf{K}_{\mathcal{N}}(\mathbf{X}, \mathbf{X})]^{\dagger}\mathbf{y}$ and $\mathcal{N}_0(\mathbf{x}') := [\phi_{\mathcal{N}}(\mathbf{x}')]^T\mathbf{w}$. As a result, the squared error can be written as follows:

$$(y^* - \mathcal{N}(\mathbf{x}'))^2 = \left( y^* - \tilde{\mathcal{N}}(\mathbf{x}') \right)^2 - 2 \cdot \left( y^* - \tilde{\mathcal{N}}(\mathbf{x}') \right) \cdot \mathcal{N}_0(\mathbf{x}') + (\mathcal{N}_0(\mathbf{x}'))^2$$

---

[1]That is, $\lim_{a \to 0} \int_{\lambda=a}^{\infty} \frac{1}{\lambda^2}\rho^{K,p}_{\mathbf{X}'}(\lambda)\mathrm{d}\lambda = \infty$.

By Lipschitz continuity of $\phi_{\mathcal{N}}$, $\mathcal{N}_0(\mathbf{x}')$ remains bounded as $\epsilon \to 0$ because $\mathbf{x}'$ is infinitesimally close to $\mathbf{x}^*$, and $\mathcal{N}_0(\mathbf{x}^*)$ is bounded by assumption on the example $(\mathbf{x}^*, y^*)$. As a result, the term that determines the divergence in the error is $\left( y^* - \tilde{\mathcal{N}}(\mathbf{x}') \right)^2$.

Let us denote $\mathbf{k}_{\mathcal{N}}(\mathbf{x}', \mathbf{X}) := [\phi_{\mathcal{N}}(\mathbf{X})]^T \phi_{\mathcal{N}}(\mathbf{x}')$. Then, the term of interest can be expanded as follows:

$$
\begin{aligned}
\left( y^* - \tilde{\mathcal{N}}(\mathbf{x}') \right)^2 = y^{*2} &- 2 \operatorname{Tr} \left( y^* \mathbf{k}_{\mathcal{N}}(\mathbf{x}', \mathbf{X}) \, \mathbf{y}^T \left( \mathbf{K}_{\mathcal{N}}(\mathbf{X}, \mathbf{X}) \right)^\dagger \right) \\
&+ \operatorname{Tr} \left( \mathbf{k}_{\mathcal{N}}(\mathbf{x}', \mathbf{X}) [\mathbf{k}_{\mathcal{N}}(\mathbf{x}', \mathbf{X})]^T \left( \mathbf{K}_{\mathcal{N}}(\mathbf{X}, \mathbf{X}) \right)^\dagger \mathbf{y} \mathbf{y}^T \left( \mathbf{K}_{\mathcal{N}}(\mathbf{X}, \mathbf{X}) \right)^\dagger \right)
\end{aligned}
\tag{3}
$$

where we have used the simple algebraic property $\mathbf{a}^T \mathbf{b} = \operatorname{Tr}(\mathbf{a}\mathbf{b}^T)$.

To conclude, let $\left( \lambda_i^{K_{\mathcal{N}}, p_{\mathbf{x}'}}, \varphi_i^{K_{\mathcal{N}}, p_{\mathbf{x}'}} \right)_{i \in \mathbb{N}}$ be the Mercer's decomposition of kernel $K_{\mathcal{N}}$ and the extended empirical distribution $p_{\mathbf{X}'}$. Then, Mercer's theorem provides an expression of the application of the kernel to data points in terms of the aforementioned eigenvalues and eigenfunctions; it follows that the empirical covariance matrix $\mathbf{K}_{\mathcal{N}}$ and the vector $\mathbf{k}_{\mathcal{N}}(\mathbf{x}', \mathbf{X})$ can be written as follows for some large enough number $M \gg P$:

$$
\mathbf{K}_{\mathcal{N}}(\mathbf{X}, \mathbf{X}) := \mathbf{\Phi} \mathbf{\Lambda} \mathbf{\Phi}^T \qquad \mathbf{k}_{\mathcal{N}}(\mathbf{x}', \mathbf{X}) = \mathbf{\Phi} \mathbf{\Lambda} \mathbf{\Phi}'
\tag{4}
$$

where $\mathbf{\Phi}_{j,k} = \varphi_k^{K_{\mathcal{N}}, p_{\mathbf{x}'}}(\mathbf{x}_j)$ for each $1 \leq j \leq P$ and $1 \leq k \leq M$, $\mathbf{\Lambda}_{k,l} = \delta_{k,l} \lambda_k^{K_{\mathcal{N}}, p_{\mathbf{x}'}}$ for each $1 \leq k, l \leq M$, and $\mathbf{\Phi}'_k := \varphi_k^{K_{\mathcal{N}}, p_{\mathbf{x}'}}(\mathbf{x}')$ for each $1 \leq k \leq M$.[2] Hence, using Mercer's decompositions, we can further expand equation 3 as follows:

$$
\begin{aligned}
\left( y^* - \tilde{\mathcal{N}}(\mathbf{x}') \right)^2 = y^{*2} &- 2 y^* \sum_j^M \frac{1}{\lambda_j^{K_{\mathcal{N}}, p_{\mathbf{x}'}}} \left( \mathbf{\Phi}^\dagger \mathbf{y} (\mathbf{k}_{\mathcal{N}}(\mathbf{x}', \mathbf{X}))^T (\mathbf{\Phi}^T)^\dagger \right)_{j,j} \\
&+ \sum_j^M \frac{1}{(\lambda_j^{K_{\mathcal{N}}, p_{\mathbf{x}'}})^2} \left( \mathbf{\Phi}^\dagger \mathbf{y} \mathbf{y}^T (\mathbf{\Phi}^T)^\dagger \right)_{j,j} \left( \mathbf{\Phi}^\dagger \mathbf{k}_{\mathcal{N}}(\mathbf{x}', \mathbf{X}) (\mathbf{k}_{\mathcal{N}}(\mathbf{x}', \mathbf{X}))^T (\mathbf{\Phi}^T)^\dagger \right)_{j,j} \\
&+ \sum_j^M \sum_{k \neq j}^M \frac{1}{(\lambda_j^{K_{\mathcal{N}}, p_{\mathbf{x}'}})(\lambda_k^{K_{\mathcal{N}}, p_{\mathbf{x}'}})} \left( \mathbf{\Phi}^\dagger \mathbf{y} \mathbf{y}^T (\mathbf{\Phi}^T)^\dagger \right)_{j,k} \left( \mathbf{\Phi}^\dagger \mathbf{k}_{\mathcal{N}}(\mathbf{x}', \mathbf{X}) (\mathbf{k}_{\mathcal{N}}(\mathbf{x}', \mathbf{X}))^T (\mathbf{\Phi}^T)^\dagger \right)_{j,k}
\end{aligned}
\tag{5}
$$

With this expression at hand, we are ready to show the statement in the theorem. For the "if" direction, assume that $\mathbf{x}'$ is $\epsilon$-adversarial, then $(y^* - \mathcal{N}(\mathbf{x}'))^2 \geq f(\epsilon)$, where $f(\epsilon) \to \infty$ as $\epsilon \to 0$

The first step is to show that as $\epsilon \to 0$, the quantities $\left( \mathbf{\Phi}^\dagger \mathbf{y} (\mathbf{k}_{\mathcal{N}}(\mathbf{x}', \mathbf{X}))^T (\mathbf{\Phi}^T)^\dagger \right)_{j,j}$, $\left( \mathbf{\Phi}^\dagger \mathbf{y} \mathbf{y}^T (\mathbf{\Phi}^T)^\dagger \right)_{j,j} \left( \mathbf{\Phi}^\dagger \mathbf{k}_{\mathcal{N}}(\mathbf{x}', \mathbf{X}) (\mathbf{k}_{\mathcal{N}}(\mathbf{x}', \mathbf{X}))^T (\mathbf{\Phi}^T)^\dagger \right)_{j,j}$ and $\left( \mathbf{\Phi}^\dagger \mathbf{y} \mathbf{y}^T (\mathbf{\Phi}^T)^\dagger \right)_{j,k} \left( \mathbf{\Phi}^\dagger \mathbf{k}_{\mathcal{N}}(\mathbf{x}', \mathbf{X}) (\mathbf{k}_{\mathcal{N}}(\mathbf{x}', \mathbf{X}))^T (\mathbf{\Phi}^T)^\dagger \right)_{j,k}$ remain bounded. Each of these quantities can be written as sums over entries of the relevant matrices. In particular, since the rectangular matrix $\mathbf{\Phi} \in \mathbb{R}^{P \times M}$ has orthogonal rows, we have $\mathbf{\Phi}^\dagger = \mathbf{\Phi}^T \left( \mathbf{\Phi} \mathbf{\Phi}^T \right)^{-1}$, and $(\mathbf{\Phi}^T)^\dagger = \left( \mathbf{\Phi} \mathbf{\Phi}^T \right)^{-1} \mathbf{\Phi}$ Moore (1920).

We have $\left| (\mathbf{k}_{\mathcal{N}}(\mathbf{x}', \mathbf{X}))_i \right| \leq \sqrt{K_{\mathcal{N}}(\mathbf{x}', \mathbf{x}') K_{\mathcal{N}}(\mathbf{x}_i, \mathbf{x}_i)}$ by Cauchy-Schwartz inequality and $K_{\mathcal{N}}(\mathbf{x}', \mathbf{x}')$ remains bounded as $\epsilon \to 0$ by Lipschitz continuity of $\phi_{\mathcal{N}}$. Entries of the matrix $\mathbf{\Phi}$ remain bounded as $\epsilon \to 0$ as evaluations of Mercer's eigenfunctions. For the same reason, entries of the matrix $\mathbf{\Phi} \mathbf{\Phi}^T$ do not diverge as $\epsilon \to 0$, and therefore neither do entries of $\left( \mathbf{\Phi} \mathbf{\Phi}^T \right)^{-1}$ by continuity of matrix inversion.

Therefore, if the squared error is to diverge towards infinity, at least one of the sums $\sum_j^M \frac{1}{\lambda_j^{K_{\mathcal{N}}, p_{\mathbf{x}'}}}$, $\sum_j^M \frac{1}{(\lambda_j^{K_{\mathcal{N}}, p_{\mathbf{x}'}})^2}$, or $\sum_j^M \sum_{k \neq j}^M \frac{1}{(\lambda_j^{K_{\mathcal{N}}, p_{\mathbf{x}'}})(\lambda_k^{K_{\mathcal{N}}, p_{\mathbf{x}'}})}$ must diverge. For small eigenvalues, the second sum dominates and hence must diverge. This sum can be expressed using the density of Mercer's

---

[2]Expression equation 4 is not the classical eigendecomposition of a square matrix: the evaluations of eigenfunctions yield rectangular (infinite) matrices. This decomposition is enabled by Mercer's theorem and is specific to kernels.

eigenvalues for $K_\mathcal{N}$ and $p_{\mathbf{X}'}$ as follows: $\lim_{M\to\infty} \sum_j^M \frac{1}{(\lambda_j^{K_\mathcal{N}, p_{\mathbf{X}'}})^2} = \int \frac{1}{\lambda^2} \rho^{K, p_{\mathbf{X}'}}(\lambda) \mathrm{d}\lambda$. Thus, the real-valued function $\lambda \mapsto \frac{1}{\lambda^2} \rho^{K, p_{\mathbf{X}'}}(\lambda)$ is not integrable near zero as required.

Conversely, assume that $\lambda \mapsto \frac{1}{\lambda^2} \rho^{K, p_{\mathbf{X}'}}(\lambda)$ is not integrable near zero. We have that, as $\epsilon \to 0$, $\left( \mathbf{\Phi}^\dagger \mathbf{y} \mathbf{y}^T (\mathbf{\Phi}^\dagger)^T \right)_{j,j}$ $\left( \mathbf{\Phi}^\dagger \mathbf{k}_\mathcal{N}(\mathbf{x}', \mathbf{X}) (\mathbf{k}_\mathcal{N}(\mathbf{x}', \mathbf{X}))^T (\mathbf{\Phi}^\dagger)^T \right)_{j,j}$ is bounded away from zero. Indeed, these terms can be written as sums of squared values, which can only converge to zero if each term of the sum converges to zero, which, in turn, only happens if $\mathbf{y} = 0$ or $\mathbf{k}_\mathcal{N}(\mathbf{x}', \mathbf{X}) = 0$ [3]. Thus, the term $\sum_j^M \frac{1}{(\lambda_j^{K_\mathcal{N}, p_{\mathbf{X}'}})^2} \left( \mathbf{\Phi}^\dagger \mathbf{y} \mathbf{y}^T (\mathbf{\Phi}^\dagger)^T \right)_{j,j} \left( \mathbf{\Phi}^\dagger \mathbf{k}_\mathcal{N}(\mathbf{x}', \mathbf{X}) (\mathbf{k}_\mathcal{N}(\mathbf{x}', \mathbf{X}))^T (\mathbf{\Phi}^\dagger)^T \right)_{j,j}$ causes a divergence in the generalisation error and example $\mathbf{x}'$ is adversarial, as required. $\square$

It follows directly from the theorem that introducing $\mathbf{x}'$ yields a Mercer's decomposition where eigenvalues have sufficient density near zero. Indeed, if eigenvalues had insufficient density near zero then the function $\frac{1}{\lambda^2} \rho^{K_\mathcal{N}, p_{\mathbf{X}'}}(\lambda)$ would be integrable near zero. In particular, according to the convergence/divergence of Riemann integrals, $\rho^{K_\mathcal{N}, p_{\mathbf{X}'}}(\lambda) \sim \lambda^\beta$ as $\lambda \to 0$ leads to non-integrability if $\beta \leq 1$ and to integrability if $\beta > 1$.

## 4 ADVERSARIAL EXAMPLES ARE EXCEEDINGLY UNLIKELY

In this section, we exploit the result in Theorem 1 to show that adversarial examples have zero measure with respect to the true data distribution $p$. As in the previous section, we assume that the neural network is trained to zero error and that its generalisation error remains bounded for all examples drawn from the same distribution $p$ as the training data. In this setting, we prove that the probability of randomly sampling an $\epsilon$-adversarial example from $p$ tends to zero as $\epsilon \to 0$. Intuitively, Theorem 1 tells us that the density function $\frac{1}{\lambda^2} \rho^{K_\mathcal{N}, p_{\mathbf{x}}}(\lambda)$ for the training data is integrable near zero; this highly restricts the probability of sampling near-zero eigenvalues, and consequently also the probability of sampling adversarial examples.

**Theorem 2.** *Let $\mathcal{N}$ be trained to zero error on $(\mathbf{X}, \mathbf{y}) \sim p$ and let $\mathbb{E}_{(\mathbf{x},y)\sim p}\left( (y - \mathcal{N}(\mathbf{x}))^2 \right) < \infty$. Consider the indicator random variable $\mathbb{1}_\epsilon^\mathcal{N}$ which determines whether a vector $\mathbf{x} \sim p$ is an $\epsilon$-adversarial example for $\mathcal{N}$ and $p$. In the limit $\epsilon \to 0$, it holds that $p(\mathbb{1}_\epsilon^\mathcal{N} = 1) = 0$.*

*Proof.* Before proving the statement of the theorem, we define a set of events with useful probabilities. For an arbitrary (but fixed) set $\mathbf{X}$ of data points sampled from $p$, let $\lambda_\mathbf{X}^j$ be the random variable assigning to each $\mathbf{x} \sim p$ the $j$-th Mercer eigenvalue for $K_\mathcal{N}$ and $p_{\mathbf{X}:\mathbf{x}}$. For an interval $[a, b]$, let $E_{[a,b]}^{\mathbf{X},j}$ be the event of $\lambda_\mathbf{X}^j$ taking values within $[a, b]$ and define the event $\{\Lambda_\mathbf{X} = \lambda\} := \bigcup_j E_{[\lambda, \lambda+\mathrm{d}\lambda]}^{\mathbf{X},j}$.

Let $\mathbb{1}_\epsilon^\mathcal{N}$ be the indicator random variable determining whether $\mathbf{x} \sim p$ is an $\epsilon$-adversarial example for $\mathcal{N}$ and $p$. By the law of total probability applied to $\mathbb{1}_\epsilon^\mathcal{N}$, the following holds:

$$p(\Lambda_\mathbf{X} = \lambda) = p(\Lambda_\mathbf{X} = \lambda | \mathbb{1}_\epsilon^\mathcal{N} = 1) \cdot p(\mathbb{1}_\epsilon^\mathcal{N} = 1) + p(\Lambda_\mathbf{X} = \lambda | \mathbb{1}_\epsilon^\mathcal{N} = 0) \cdot p(\mathbb{1}_\epsilon^\mathcal{N} = 0) \quad (6)$$

The probability $p(\Lambda_\mathbf{X} = \lambda)$ coincides with the average density of Mercer's eigenvalues with respect to the random vector $\mathbf{x} \sim p$:

$$p(\Lambda_\mathbf{X} = \lambda) = \int p(\Lambda_\mathbf{X} = \lambda | \mathbf{x}) p(\mathbf{x}) \mathrm{d}\mathbf{x} = \int \rho^{K_\mathcal{N}, p_{\mathbf{X}:\mathbf{x}}}(\lambda) p(\mathbf{x}) \mathrm{d}\mathbf{x} \quad (7)$$

Since $\mathbb{E}_{(\mathbf{x},y)\sim p(\mathbf{x},y)}\left( (y - \mathcal{N}(\mathbf{x}))^2 \right) < \infty$, it is easy to show, using the same arguments as in the proof of Theorem 1, that $\lambda \mapsto \frac{1}{\lambda^2} p(\Lambda_\mathbf{X} = \lambda)$ is integrable near zero.

By definition, $p(\Lambda_\mathbf{X} = \lambda | \mathbb{1}_\epsilon^\mathcal{N} = 1)$ is the average density of Mercer eigenvalues for $K_\mathcal{N}$ and $p_{\mathbf{X}:\mathbf{x}}$ given that $\mathbf{x}$ is an $\epsilon$-adversarial example. By Theorem 1, in the limit $\epsilon \to 0$, $p(\Lambda_\mathbf{X} = \lambda | \mathbb{1}_\epsilon^\mathcal{N} = 1)$ is strictly positive near zero, which gives us the following identity as $\lambda \to 0$ and $\epsilon \to 0$:

$$p(\mathbb{1}_\epsilon^\mathcal{N} = 1) = \frac{p(\Lambda_\mathbf{X} = \lambda)}{p(\Lambda_\mathbf{X} = \lambda | \mathbb{1}_\epsilon^\mathcal{N} = 1)} - \frac{p(\Lambda_\mathbf{X} = \lambda | \mathbb{1}_\epsilon^\mathcal{N} = 0)}{p(\Lambda_\mathbf{X} = \lambda | \mathbb{1}_\epsilon^\mathcal{N} = 1)} \cdot p(\mathbb{1}_\epsilon^\mathcal{N} = 0) \quad (8)$$

---

[3]Note that these excluded edge cases are already taken in account as particular cases of equation 2.

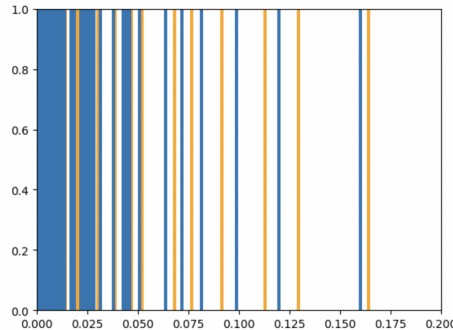

Figure 1: Eigenvalue distribution for kernel $K_\mathcal{N}$ and the original data distribution (respectively, the the same data distribution extended with one adversarial example) in orange (respectively, in blue). The X-axis is indexed by the value of the eigenvalues and the Y-axis is a probability density.

By Theorem 1, $\lambda \mapsto \frac{1}{\lambda^2} p(\Lambda_\mathbf{X} = \lambda | \mathbb{1}_\epsilon^\mathcal{N} = 1)$ is not integrable near-zero whereas $\lambda \mapsto \frac{1}{\lambda^2} p(\Lambda_\mathbf{X} = \lambda | \mathbb{1}_\epsilon^\mathcal{N} = 0)$ is integrable near-zero. In particular, $\frac{p(\Lambda_\mathbf{X} = \lambda | \mathbb{1}_\epsilon^\mathcal{N} = 0)}{p(\Lambda_\mathbf{X} = \lambda | \mathbb{1}_\epsilon^\mathcal{N} = 1)} \to 0$ as $\lambda \to 0$. Similarly, $\frac{p(\Lambda_\mathbf{X} = \lambda)}{p(\Lambda_\mathbf{X} = \lambda | \mathbb{1}_\epsilon^\mathcal{N} = 1)} \to 0$ as $\lambda \to 0$, which yields $p(\mathbb{1}_\epsilon^\mathcal{N} = 1) = 0$, as required. □

Our findings indicate that in practical evaluations of neural models, it is highly improbable for test sets to include adversarial examples, given that they are sampled from the same underlying distribution $p$ as the training data. In essence, adversarial examples created through artificial perturbations of samples drawn from $p$ are considered out-of-distribution and thus extremely unlikely to originate from the same underlying data generation process that produced the training and test data. This substantiates the intuition that adversarial examples do not naturally manifest in real-world scenarios.

## 5 EXPERIMENTS

To validate our theory, we conducted experiments on a subset of MNIST consisting exclusively of classes "0" and "1." This subset comprised 253 examples, each with 784 pixels per image. Additionally, we established a test set consisting of 17 examples. We deliberately opted for this reduced dataset size to accommodate computational constraints, as our computations require diagonalising kernel matrices with dimensions of $(P + 1) \times (P + 1)$. However, it is important to note that the size of the dataset is inconsequential to our theoretical results, as they remain independent of dataset scale.

All experiments were executed on a GPU-enabled platform within Google Colab for enhanced computational efficiency.

Using Pytorch, we trained a ReLU FCN with one hidden layer of size $N = 512$ to zero error on our training dataset and $100\%$ accuracy on our restricted test set. Subsequently, we exploited the implementation of the DeepFool algorithm Moosavi-Dezfooli et al. (2016) to generate one adversarial example for each of the 17 test examples. This algorithm essentially involves an iterative process wherein we continuously adjust the input in the direction of the normalised gradient until a change in prediction occurs.

We computed the eigenvalue distributions of the empirical kernel for the original training data distribution. Then, for each adversarial example, we computed the updated eigenvalue distribution for the same empirical kernel and the training data extended with the adversarial example. Thus, this gives us 17 different updated eigenvalue distributions. To compute these distributions, we constructed 17 new kernel matrices by adding the relevant row and column corresponding to the adversarial example and diagonalised them using PyTorch. By the classical results of Baker (1977); Rasmussen & Williams (2006), the largest Mercer eigenvalues roughly coincide with the eigenvalues obtained by diagonalising the corresponding kernel matrix. We have plotted an example of such updated distributions against the original one in Figure 1; note how introducing an adversarial example shifts the eigenvalue distribution towards zero.

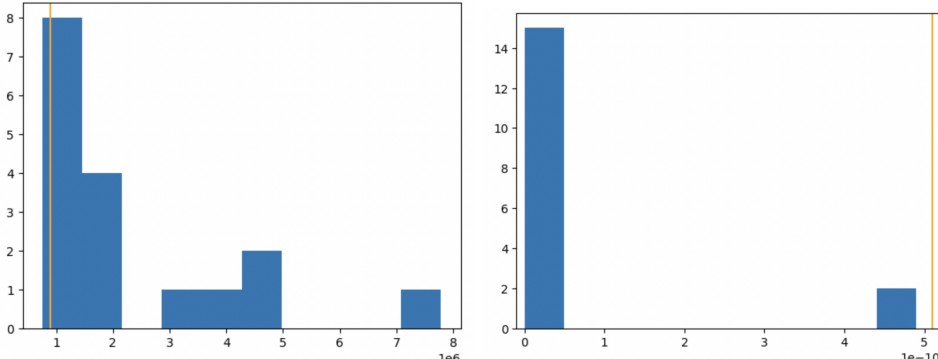

Figure 2: Comparing the integral of interest (left subfigure) and the minimal eigenvalue (right sub-figure) between the original data distribution (orange line) and the updated data distributions (in blue). The X-axis is indexed by the value of the integral quantity (respectively, the minimal value of eigenvalues) on the left (respectively, on the right). In turn, the Y-axis is indexed on the left (respectively, on the right) by the number of updated data distributions whose estimated integral (respectively, whose minimal eigenvalue) falls into that bin.

We then computed the minimal eigenvalue within each eigenvalue distribution, and we estimated the integral near zero of our real-valued function $\lambda \mapsto \frac{1}{\lambda^2}\rho(\lambda)$ by computing the sum $\frac{1}{B}\sum_j \frac{1}{\lambda_j^2}\rho(\lambda_j)$ for the relevant densities of Mercer eigenvalues over the $B$ bins in the histogram of Figure 1. The histogram representing these estimations is depicted in Figure 2. Our numerical results clearly demonstrate that, as expected, introducing adversarial examples shifts the eigenvalue distribution towards zero and inflates the value of the integral of interest.

## 6    RELATED WORK

Several explanations have been proposed for the pervasiveness of adversarial attacks in neural networks. The *linearity hypothesis* Goodfellow et al. (2015) posits that, because of the (local) linear nature of trained neural networks, small changes to each component of a high-dimensional vector amount to a large change in the network's output. The universality of $\ell_2$-adversarial attacks on ReLU networks with random weights Daniely & Shacham (2020) provides a strong argument supporting this hypothesis, especially considering the locally linear characteristics of ReLU networks.

The linearity hypothesis has motivated a series of research endeavors analysing the topological characteristics of decision boundaries, with the aim to elucidate the nature of adversarial examples and develop techniques for enhancing adversarial robustness. In Tanay & Griffin (2016), adversarial examples are attributed to the distance of the sampling subspace to the decision boundary. These potential deficiencies in the topology become more pronounced in higher dimensions, a phenomenon exacerbated by the curse of dimensionality. Notably, the susceptibility to adversarial attacks escalates with the increase in input dimensionality, as observed in Simon-Gabriel et al. (2019). In some instances, particularly for synthetic data distributions with sufficiently high dimensions, adversarial attacks become nearly unavoidable, as noted in Shafahi et al. (2019). Moreover, data sparsity in relation to the input space heightens vulnerability to adversarial attacks, as discussed in Paknezhad et al. (2022); Weitzner & Giryes (2023). The dimensionality of the parameter space also plays a significant role in this context, with parameter redundancy Paknezhad et al. (2022) and a large $\ell_1$-norm of the parameters Guo et al. (2018) representing situations associated with increased vulnerability.

Another line of research identifies features as the key object driving the occurrence of adversarial examples. In Ilyas et al. (2019), it was shown that *non-robust features*—features resulting from spurious correlations within the data that are nevertheless highly predictive— are responsible for the presence of adversarial examples. Similarly, Wang et al. (2017) established that the existence of an unnecessary feature, introduced to replicate the true underlying target function, renders a system vulnerable to adversarial attacks. This perspective is further supported by the observation that

saliency methods tend to emphasise class-discriminative features that can be exploited to generate adversarial attacks, as highlighted in Gu & Tresp (2019).

We argue that our approach is positioned at the intersection of both these streams of research. By taking Mercer's eigenvalues into account, our theory operates in the high-dimensional embedding space where the predictions follow a linear pattern. Additionally, shifting towards zero eigenvalues can be understood as exploiting directions of non-robust features. Indeed, our approach was first inspired by the double-descent phenomenon in neural networks Mei & Montanari (2022), in which the spectrum of the empirical kernel also shifts towards zero at the divergence in terms of generalisation error. Within this body of literature, it is already established that stochastic cancellations give rise to the emergence of spurious directions within the feature space Harzli et al. (2023), rendering them susceptible to overfitting.

A final perspective on adversarial attacks, which bears loose connections with our approach, is based on information geometry Zhao et al. (2019); Naddeo et al. (2022). This perspective centers on the use of the *Fisher Information Matrix* which quantifies, for any pair of data points, the correlation between gradients (with respect to parameters) of the log likelihood of the data. Researchers have demonstrated that a technique involving the iterative elimination of the dominant eigenvalue direction in the Fisher Information Matrix leads to the generation of adversarial examples. This process appears to yield configurations where variations in network parameters exhibit strong linear dependencies with respect to the data. Consequently, these configurations may manifest as scenarios where certain features become perfectly correlated within the dataset and this, in turn, results in the empirical kernel having Mercer eigenvalues that approach zero.

## 7 LIMITATIONS AND FUTURE WORK

Our paper provides mathematical underpinning for a principled understanding of adversarial examples and sheds light on the mounting empirical evidence that adversarial examples exhibit measure zero with respect to the data distribution.

Our results, however, are currently only directly applicable to FCNs and regression tasks. Nevertheless, we anticipate that the core insights from our research could be extended to encompass classification tasks as well as other neural architectures; this is a clear path for future research. Additionally, conducting comprehensive experiments on more extensive and diverse datasets would be beneficial. It is worth noting that our results can be readily extended to scenarios where the feature map exhibits similar desirable properties as those seen in FCNs, such as Lipschitz continuity. Importantly, our findings remain unaffected by variations in problem dimensions and data characteristics, offering a high degree of versatility.

It is also important to emphasise that our results are derived in the limit where adversarial examples become infinitesimally close to data points sampled from the original data distribution. Specifically, we have not characterised the rate at which the probability of encountering adversarial examples diminishes with the distance to an example from the true data distribution. Addressing this aspect remains a subject for future research and requires the derivation of precise bounds for the generalisation error. Exploring specific criteria governing the data distribution and neural architecture that lead to rapid convergence rates (as $\epsilon \to 0$) and result in low probabilities of generating adversarial examples is a promising avenue. Achieving this understanding could lay the groundwork for designing architectures aimed at enhancing robustness, which, in turn, may lead to improved generalisation performance.

Additionally, while we have proved that adversarial examples have a measure zero with respect to the data distribution in the limit, we have not shown that examples producing near-zero Mercer eigenvalues for the empirical kernel of a FCN always exist, which would be consistent with existing empirical evidence and theoretical results.

In conclusion, we anticipate that our findings will serve as a catalyst for additional research into the characteristics and prevalence of adversarial examples. Furthermore, we hope they will also encourage the exploration and development of novel defense mechanisms against adversarial attacks.

**Reproducibility statement.**  The full proofs of the theorems are included in the main paper and the code for numerical experiments is part of the supplementary material.

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
