# OpenReview forum: "Adversarial Attacks as Near-Zero Eigenvalues in The Empirical Kernel of Neural Networks"
_ICLR.cc/2024/Conference — Submitted to ICLR 2024_

### Official Review · Reviewer_YsYz · 2023-10-29

**Soundness:** 3 good
**Presentation:** 3 good
**Contribution:** 3 good
**Rating:** 6
**Confidence:** 3

**Summary:**

This paper makes contributions to the theoretical understanding of the adversarial examples. The authors use kernel theory, Mercer’s theorem in particular, to explain the adversarial examples. Specifically, the authors first prove that adversarial examples shift the Mercer’s spectrum of the empirical kernel so that the near-zero density of the Mercer eigenvalues is high. Extending the first theorem, the authors also explain the reason for the denseness of adversarial examples, i.e., adversarial examples are unlikely to appear in the test set, but they exist near every test sample. Lastly, the authors demonstrate their findings with small experiments.

**Strengths:**

1. I briefly checked the proofs and the proofs look to be correct to me.
2. To the best of my knowledge, this is an original work that makes theoretical progress in the field of adversarial machine learning. Considering that such theoretical works are rare in this field, the authors’ findings are valuable.

**Weaknesses:**

1. As mentioned in Section 7, the theoretical result only covers limited neural network architecture, i.e., fully connected layers.
2. This work is a theory-intensive paper, however, experiments can be improved further.
    - For example, the authors can run more experiments on artificial data to reduce the heavy computation of diagonalization and then validate the theory more thoroughly.
    - DeepFool algorithm is an old attack algorithm and cannot represent all the existing attacks (that are likely to be more powerful than DeepFool). The authors should perform similar experiments with other attack methods.

**Questions:**

1. I cannot understand the reason for the assumption on the layer widths, i.e., all layers except the last layer have the same width $N$, because the proof mainly uses the empirical feature map that does not involve the intermediate layer output. Is this assumption necessary for the theorem? If so, how do you justify the assumption on the layer width?
2. Minor comments
    - According to the [formatting instruction](https://github.com/ICLR/Master-Template/raw/master/iclr2024.zip), in-text citations (`\citet`) and citations in parentheses (`\citep`) should be used differently, but I see only in-text citations in the paper writing. Please fix the citation style.
    - I don’t think that Figure 1 is the best way to show the eigenvalue distributions. Why do you waste space by having a y-axis ranging from 0 to 1?

---

> ### Author Response · Authors · 2023-11-13
>
> Remark: As mentioned in Section 7, the theoretical result only covers limited neural network architecture, i.e., fully connected layers.
>
> Answer: As pointed out in the response to Reviewer DyBR and as indicated also in Section 7, our results can be seamlessly extended to any architecture where the feature map exhibits Lipschitz continuity.
>
> Remark: DeepFool algorithm is an old attack algorithm and cannot represent all the existing attacks (that are likely to be more powerful than DeepFool). The authors should perform similar experiments with other attack methods.
>
> Answer: As pointed out in Section 7, additional experiments would indeed be beneficial. Please note, however, that our theory is independent from the characteristics of the data as well as of the adversarial attack method. We chose DeepFool as a widely-known algorithm simply for illustrative purposes. The power of the attack method does not affect our theory: as long as one is able to find an adversarial example for the task at hand, it can be described by our framework.
>
> Question: I cannot understand the reason for the assumption on the layer widths, i.e., all layers except the last layer have the same width $N$, because the proof mainly uses the empirical feature map that does not involve the intermediate layer output. Is this assumption necessary for the theorem? If so, how do you justify the assumption on the layer width?
>
> Answer: You are correct in noticing this unnecessary simplification, which was in place only to avoid
> cumbersome notation. As already mentioned, our theory applies to any neural architecture where the feature
> map exhibits Lipschitz continuity.

---

### Official Review · Reviewer_74hn · 2023-10-30

**Soundness:** 3 good
**Presentation:** 3 good
**Contribution:** 3 good
**Rating:** 5
**Confidence:** 3

**Summary:**

The paper proposes a possible explanation for the existence of adversarial examples which allows for the emergence of the property that adversarial examples are hard to find (or does not occur naturally in test time).

As for existence, the paper proposes that an adversarial example is a point $x'$ in an $\epsilon$-neighborhood of a natural sample $x^*$ for which the squared error $(y(x^*)-y(x'))^2$ diverges as $\epsilon\to 0$. The paper then proves that, for this definition of an adversarial example, the eigenvalues of the eigenfunctions of the Mercer's kernel vanish. Equivalently, $\lim_{t\to0}\int_t^\infty\frac{1}{\lambda^2}\mathrm{d}\mu(\lambda)=\infty$. Next, the previous theorem is utilized to show that the measure of adversarial examples vanishes as $\epsilon\to0$.

Given the computational constraints, the predictions of the theorems are verified empirically by considering a binary classification of 0 and 1 digits in the MNIST dataset.

**Strengths:**

### Originality:
Even though some aspects of the proposal appear in the literature, the paper adds an original perspective on the existence of adversarial examples from the point of view of kernels.

### Quality:
The paper is exceptionally well-written. The assumptions and definitions are formally expressed, and claims of the paper is presented in concise steps and a rigorous manner.

### Clarity:
The paper is clear in its goals and provides enough background for the average reader to understand the logic behind the claims.

### Significance:
The proposed theorems could prove to be consequential in interpretation and mitigation of adversarial examples phenomenon.

**Weaknesses:**

### Minor:
- The axis of the figures does not bear any labels, and the captions are also a little encrypted.

- The low-probability pockets perspective is credited to Goodfellow in the abstract, which is wrong. The main text however correctly credits Szegedy.

### Major:
- My main objection with the presented argument is that I am not sold on the definition of an adversarial example in the paper. Specifically, I think that the limiting process of $\epsilon\to0$ in the definition is flawed. This limiting process effectively adds two equal samples to the training set with two different targets. I am under the impression that the vanishing of the eigenvalues is a consequence of this construction ($K(X,X)$ wouldn't be invertible) and is not associated with the robustness of the network.

- Assuming that my understanding of the limiting process is correct, the paper has rediscovered the robustness-accuracy trade-off as depicted in [A] (this is not a critique exactly). Zhang et al. in [A] propose that adversarial examples and natural samples overlap in the input space and that is why we observe a trade-off between robustness and accuracy in adversarial training. I think the paper should at the very least mention [A]. An alternative to the proposal of Zhang et al. that might be relevant is [B].

[A]: https://proceedings.mlr.press/v97/zhang19p.html
[B]: https://arxiv.org/abs/2309.17048

**Questions:**

- The limiting process that constructs $\int_t^\infty\frac{1}{\lambda^2}\mathrm{d}\mu(\lambda)$ appear to be a Riemannian sum. However, I am curious to know if $M\to\infty$ is the same as asserting that the size of the training set approaches infinity.

---

> ### Author Response · Authors · 2023-11-13
>
> Remark: The limiting process of $\epsilon \to 0$ in the definition of an adversarial example seems flawed.
>
> Answer: Please check our general remark to all reviewers and the corresponding modifications in the paper. Do not hesitate to ask should these modifications not adequately address your comments.
>
> Remark: Assuming that my understanding of the limiting process is correct, the paper has rediscovered the robustness-accuracy trade-off as depicted in A (this is not a critique exactly). Zhang et al. in A propose that adversarial examples and natural samples overlap in the input space and that is why we observe a trade-off between robustness and accuracy in adversarial training. I think the paper should at the very least mention A. An alternative to the proposal of Zhang et al. that might be relevant is B.
>
> Answer: Thank you for pointing out these references. These potential connections are indeed interesting and worth exploring. Please note, however, that the focus of our work differs from that by Zhang et al., as we give a mathematical justification as to  why adversarial examples are rarely sampled by chance in practice, even if they "overlap with natural samples".
>
> Question: The limiting process that constructs $\int_t^\infty \frac{1}{\lambda^2} \mathrm{d} \mu (\lambda)$ appear to be a Riemannian sum. However, I am curious to know if $M \to \infty$ is the same as asserting that the size of the training set approaches infinity.
>
> Answer: Here, $M \to \infty$ refers to the infinite sum in the Mercer's decomposition theorem. This infinite sum is well-defined regardless of the size of the dataset.

---

> > ### Comment · Reviewer_74hn · 2023-11-22
> >
> > Thank you for the response, but I do not find the general remarks very relevant to my question. Similar to other reviewers, I think that the papers perspective is promising. However I am under the impression that the assumptions in the proof theorem 1 are not warranted, and need some better argumentation, specially regarding infinities.
> >
> > In the proof of theorem 1, the paper assumes that $(\Phi(X)\Phi(X)^T)^{-1}$ is bounded, but my understanding is that the assumption of $\epsilon\to0$ and $(y-N(x'))^2\to\infty$ is the same as assuming $\Phi(x'\cup X)\Phi(x'\cup X)^T$ is not invertible (because there is some $x$ in the dataset that has been added to the set twice, but with different targets). Consequently, generalization error is not the source of the error that is being analyzed in the paper.

---

> > > ### Author Response · Authors · 2023-11-22
> > >
> > > The assumptions of $\epsilon \to 0$ "and $y - \mathcal{N} (x'))^2 \to \infty$ are not the same as assuming $K (X',X')$ i not invertible. There is no $x$ in the dataset that has been added to the set twice with different targets. Instead, the limit induces an example which is very close to an arbitrary test example but for which the predictions remains apart from one another (which is the very intuition of adversarial behavior). Adding this example adds one row and one column (the kernel evaluations between the new point and each of the training data points) to the matrix $K (X, X)$. It is perfectly fine to assume that $(\phi (X) \phi (X)^T)^{-1}$ is bounded, as we do not modify the training set.
> > >
> > > As one can see in our proof of Theorem 1, the main insight of the proof is that introducing an adversarial example creates a divergence caused by the pseudo-inverse of $K (X', X')$ (due to eigenvalues becoming close to zero) so it is in a sense making the matrix $K (X', X')$ "almost not invertible".

---

### Official Review · Reviewer_vXdv · 2023-11-05

**Soundness:** 2 fair
**Presentation:** 1 poor
**Contribution:** 2 fair
**Rating:** 5
**Confidence:** 3

**Summary:**

The paper studies adversarial examples in the context of deep supervised learning and aims to show that adversarial examples are of low probability. The paper introduces a kernel-based framework to analyze adversarial examples, connecting adversarial examples to minor Mercer’s eigenvalues in the empirical kernel matrix of the neural net. Based on a definition of adversarial examples in section 2, the paper proves Theorem 1 showing that when the attack radius $\epsilon$ is approaching $0$ the existence of an adversarial example means the density function of the kernel eigenvalues divided by $\lambda^2$ is not integrable and uses this result to show in the limit case $\epsilon\rightarrow 0$ the adversarial examples have zero probability. Some numerical results on MNIST data and DeepFool attacks are presented in the paper.

**Strengths:**

1- The paper applies a kernel-based framework to analyze adversarial examples which I find interesting. I appreciate the authors' idea of connecting the Mercer eigenvalues to adversarial examples.

**Weaknesses:**

1- The paper's presentation should be improved. The theory sections are abstract and hard to follow in their current form. I think this is because the authors present the results in the most general and abstract way possible and also include the theorem proofs in the text, which makes the paper difficult to read for an average machine learning researcher. I suggest the authors postpone the proofs to an appendix and discuss some corollaries and examples of the theorems for some basic kernel functions.

2- Theorem 1 is an asymptotic result and holds when the attack radius $\epsilon\rightarrow 0$. I think this could limit the implications of the theorem. Also, it seems that in Theorem 1 the bound on $\epsilon$ in the limit statement depends on the choice of adversarial example $x'$ which means the asymptotic guarantee would not uniformly hold for all $\epsilon\le \epsilon_0$ with $\epsilon_0$ being independent of adversarial example $x'$.

3- The definitions and notations in sections 2 and 3 are in some cases vague and raise questions: a) In the definition of adversarial examples, what is function $f(\epsilon)$? The definition does not specify function $f$ and only states the condition $\lim_{\epsilon\rightarrow 0}f(\epsilon) = \infty$, which would be problematic because for a fixed $\epsilon >0$  we can always find a function $f$ for observed data that makes the definition hold for a perturbed $x'$. Also, in the proofs of Theorem 1,2 do the authors determine the function $f$ based on the dataset $X, y$ or is the choice of $f$ independent of $X,y$?

b) In Theorem 1, what is the definition of $P_{X'}$ (with the prime)? Is it the delta function on $x'$ or a continuous density function related to data distribution $P_X$?

c) I wonder what the authors precisely mean when they say "In the limit $\epsilon\rightarrow 0$, .... " in Theorems 1,2. Does that mean there is an $\epsilon_0 >0$ independent of the choice of $x'$ such that the statement holds for every $\epsilon\le \epsilon_0$? (clarification on  weakness 2)

d) As another question, what is the precise mathematical definition of "such that $|| x- x'||\le \epsilon$ for some example $(x^*,y)\sim p$"? If $p$ is a continuous distribution, then every point in $p$'s sample space has zero probability so whether $(x^*,y)$ is in the support set of $p$ or not does not change its zero likelihood to be sampled from $p$. The theorem should explain this sentence.

**Questions:**

Please see the questions in the previous part.

---

> ### Author Response · Authors · 2023-11-13
>
> Remark:
> It seems that in Theorem 1 the bound on $\epsilon$ in the limit statement depends on the choice of adversarial example $\mathbf{x}'$ which means the asymptotic guarantee would not uniformly hold for all $\epsilon \leq \epsilon_0$ with $\epsilon_0$ being independent of adversarial example.
>
> Answer:
> In Theorem 1, the possible adversarial examples $x'$ depend on the choice of $\epsilon$, not the other way around. Hence the asymptotic guarantee does hold uniformly for all $\epsilon \leq \epsilon_0$ with $\epsilon_0$ independent of the adversarial example.
>
>
> Questions: a) In the definition of adversarial examples, what is function $f (\epsilon)$? The definition does not specify function $f$ and only states the condition $\lim_{\epsilon \to 0} f (\epsilon) = \infty$, which would be problematic because for a fixed $\epsilon >0$ we can always find a function $f$ for observed data that makes the definition hold for a perturbed $\mathbf{x}'$. Also, in the proofs of Theorem 1,2 do the authors determine the function $f$ based on the dataset $\mathbf{X}, \mathbf{y}$ or is the choice of $f$ independent of $\mathbf{X}, \mathbf{y}$?
>
> Answer:  Please check our general remark to all reviewers and the corresponding modifications in the paper. Do not hesitate to ask should these modifications not adequately address your comments.
>
>
>
> Question: In Theorem 1, what is the definition of $p_{\mathbf{X}'}$ (with the prime)? Is it the delta function on $\mathbf{x}'$ or a continuous density function related to data distribution $p_{\mathbf{X}'}$?
>
> Answer: The definition of \emph{empirical probability distribution} for a dataset is provided in Section 2 at the end of the paragraph on empirical feature maps; it is essentially a summation of Dirac deltas. It's important to note that this definition is generic, and applicable to any design matrix (in particular to $\mathbf{X}')$. The notation $\mathbf{X}' = \mathbf{X} : \mathbf{x}'$ represents the concatenation of matrix $\mathbf{X}$ and vector $\mathbf{x}'$, as described also in Section 2.
>
>
> Question: I wonder what the authors precisely mean when they say "In the limit $\epsilon \to 0$, .... " in Theorems 1,2. Does that mean there is an $\epsilon_0 > 0$ independent of the choice of $\mathbf{x}'$ such that the statement holds for every $\epsilon \leq \epsilon_0$?
>
> Answer: It implies the existence of an $\epsilon_0$ such that the statement holds for all $0 <\epsilon \leq \epsilon_0$. The possible choices for $\mathbf{x}'$ depend on $\epsilon$, but not the other way around.
>
> Question: What is the mathematical definition of "such that $|| \mathbf{x}^* - \mathbf{x}' || \leq \epsilon$ for some example $(\mathbf{x}^*, y^*) \sim p$.
> "? If $p$ is a continuous distribution, then every point in $p$'s sample space has zero probability so whether $(\mathbf{x}^*, y^*)$ is in the support set of $p$ or not does not change its zero likelihood to be sampled from $p$.
>
> Answer: Each individual point has probability zero but not a zero \emph{probability density}.
>  For instance, in a Gaussian density, each real number has a probability of zero when sampled as a specific event; however, it's essential to note that each real number does have a non-zero probability density.
> The statement
> $(\mathbf{x}^*, y^*) \sim p$
> simply means that point $(\mathbf{x}^*, y^*)$ is sampled according to the same joint probability density $p$ as that of the training data.

---

### Official Review · Reviewer_DyBR · 2023-11-08

**Soundness:** 2 fair
**Presentation:** 3 good
**Contribution:** 2 fair
**Rating:** 6
**Confidence:** 2

**Summary:**

This paper studies a novel interpretations of adversarial attacks as points that induce near zero Mercer eigenvalues in the kernel formed via the inner product of empirical feature maps (i.e. the activations at the penultimate network layer). The authors show under strong assumptions of locality that the set of adversarial examples is measure zero under the training data distribution, explaining why such points are practically never observed in the natural world. Preliminary experiments support the theoretical claims.

This reviewer was admittedly unable to check the details of the proofs to ensure correctness.

**Strengths:**

This paper proposes a new mathematical explanation for the existence of adversarial examples through the application of kernel methods. Most hypotheses to date about this phenomena have lacked formal proofs. A successful mathematical model for adversarial examples could have significant impact by informing techniques to improve adversarial robustness.

The paper explains that neural networks generalize in the real world because the set of adversarial examples is measure zero. This proof supports a decade-old conjecture by Goodfellow et al.

The paper does an excellent job framing the challenge and decade-long history of adversarial examples. Notation is nicely defined, consistent, and clear throughout the paper.

**Weaknesses:**

The paper is narrow in scope and parts of the paper appear to be hastily written. It is unclear whether the contribution is complete enough to justify a top-tier publication. In particular, the fact that the paper limits its consideration to adversarial points that are infinitesimally close to data points is probably unrealistic.

The definition of adversarial examples appears a bit different than the usual definitions, relying on a local limiting assumption. Some work needs to be done to explain the relationship between this paper’s definition and the working definition used of the broader community.

The figures are poorly presented, and captions could be clarified.

The paper says “Our results, however, are currently only directly applicable to FCNs and regression tasks. Nevertheless, we anticipate that the core insights from our research could be extended to encompass classification tasks as well as other neural architectures; this is a clear path for future research.” It is not clear in the experiments that the authors are training regression tasks on MNIST, if this is the case, this needs to be clarified in the paper.

Small issues:
* Please check for style, e.g. correct use of \citet and \citep.
* At the end of Section 1, the authors say they “estimated the integral of relevant quantities near zero.” Please make this more specific to outline contributions up front.
* Space before comma 2nd line from the bottom on page 1.
* At the end of Section 6, add a citation for the claim “Researchers have demonstrated that a technique involving the iterative elimination of the dominant eigenvalue direction in the Fisher Information Matrix leads to the generation of adversarial examples.”

**Questions:**

1. Are there geometric interpretations that your theory provides?
2. Are the authors proposing that this method could serve to identify adversarial examples? If so, is there any contention with [1]?
3. The notation $(x^*, y^*) \sim p$ is strange from a probabilistic perspective. Isn’t there some probability mass on any point $x^*$? It also appears you are assuming no label noise (i.e. the labeling function $y(x)$ is deterministic).
4. Is it really necessary to limit this paper to the consideration of fully-connected layers, or is a Lipschitz assumption sufficient?

[1] Tramer, Florian. “Detecting adversarial examples is (nearly) as hard as classifying them." In International Conference on Machine Learning. 2022.

---

> ### Author Response · Authors · 2023-11-13
>
> Remark: The definition of adversarial examples appears a bit different than the usual definitions, relying on a local limiting assumption. Some work needs to be done to explain the relationship between this paper’s definition and the working definition used of the broader community.
>
> Answer:  Please check our general remark to all reviewers and the corresponding modifications in the paper. Do not hesitate to ask should these modifications not adequately address your comments.
>
> Question: Are there geometric interpretations that your theory provides?
>
> Answer: We briefly mentioned the geometric interpretation of our results in Section 6, paragraph 4. We indicated the relationship between the shift towards zero eigenvalues and the shift in the input feature space towards the direction of non-robust features.
> The latter is a common observation in the literature on adversarial attacks.
>
> Question: Are the authors proposing that this method could serve to identify adversarial examples? If so, is there any contention with [1]?
>
> Answer: In their current form, our results may not be directly applicable for detecting adversarial examples in practical settings. Theorem 1 relies on an abstract test example that the neural network generalizes well to, and such examples are not always accessible in real-world scenarios.
>
> Question: The notation $(x^* , y^*) \sim p$  is strange from a probabilistic perspective. Isn’t there some probability mass on any point $(x^* , y^*)$? It also appears you are assuming no label noise (i.e. the labeling function $y(x)$ is deterministic).
>
> Answer: This notation indicates that $(x^*, y^*)$ is a data point sampled from the same distribution as the training data, with joint probability density $p$.
> As stated by the reviewer, any point $(x^*, y^*)$
> holds a probability mass, and it's precisely with respect to that probability measure that we showed the measure-zero property of adversarial examples.
> Since $p (x^*, y^*) $denotes a joint probability density, noise can be easily incorporated in its definition.
>
> Question: Is it really necessary to limit this paper to the consideration of fully-connected layers, or is a Lipschitz assumption sufficient?
>
> Answer: Your observation is accurate. As pointed out in the second paragraph of Section 7, our results can be easily  extended to encompass other architectures verifying Lipschitz continuity.

---

> > ### Comment · Reviewer_DyBR · 2023-11-20
> > **Thank you for modifying the paper, slightly raising my score**
> >
> > I appreciate the authors' detailed rebuttal, and especially appreciate the edits made in the paper to clarify the definition of adversarial examples. On the negative side, I tend to agree with Reviewer YsYz that the experiments are currently a bare minimum. On the positive side, the paper introduces a nice framework in which to analyze adversarial examples (through the lens of the empirical kernel).
> >
> > I am willing to slightly raise my score, and encourage the authors to continue to refine the draft if the paper is not accepted.

---

### Author Response · Authors · 2023-11-13
**General responses for all reviewers**

We thank all reviewers for their comments.
Reviewers found our definition of an adversarial example in Section 2 confusing, and we apologise for this. We have rewritten the definition
and added additional explanations
to make the key points clearer. None of our technical results is affected by the changes in the definition.

As described in the revised submission, function f is introduced in the definition of adversarial examples to adjust the standard
notions in the literature for classification to the regression setting.
In the context of regression there is no a-priori notion of what it means for the adversarial example to ``change" the prediction (in classification, this is obvious because changing the prediction means predicting a different class). Therefore, function f allows practitioners to quantify in a task-dependent way what it means for the modification of the output to be significant enough. This function may be task-dependent, but it does not depend on the observed data; hence, in the proofs of Theorems 1 and 2, f is independent of X and y. To make this clear, in the revised definition we no longer quantify existentially over f, but rather fix it beforehand to reflect that it captures a choice by the practitioner.

Some of the reviewers also commented on the fact that the statements in Theorems 1 and 2 are asymptotic, and this may limit their applicability.
Although this was acknowledged in Section 7, we would like to emphasise that a limit to small epsilon is
somewhat built into the intuition behind adversarial examples as those leading to a significant change in the output despite being extremely close to a data point sampled from the original data distribution. Indeed, for values of epsilon that are not small, it becomes trivial to find examples altering the output of the model in a substantial way.

---

### Meta-Review · Area_Chair_sm8M · 2023-12-25

**Metareview:**

This paper offers an interpretation of adversarial attacks, suggesting they correspond to instances that lead to nearly zero Mercer eigenvalues in the kernel of neural network feature maps. The paper then argues that this interpretation can explain, under certain assumptions, that adversarial examples have a measure zero in the training data distribution, which accounts for their rarity in natural data. The paper supports the theoretical findings with preliminary experiments on MNIST using DeepFool attacks.

While reviewers appreciate the new theoretical interpretation of adversarial examples, they find the theoretical statements and definitions confusing. In particular, various reviewers point out that the statement "in the limit epsilon --> 0" is unclear. I read Theorem 1 and also find the statement confusing. It would be much cleaner if Theorem 1 is stated in a standard mathematical way (using the standard language of "existence" and "for all", instead of the informal language "in the limit epsilon -->0"). Besides the formal statement, the authors can provide an additional informal statement of Theorem 1 to increase the readability, which is a common practice in machine learning papers.

I also find the definition of adversarial examples confusing, and there are other questions that can be asked for this definition besides the questions by reviewers. For instance, I think the authors expected the good prediction error (y* - N(x*))^2 < (y* - N(x'))^2 which is the adversarial prediction error,  but based on the definition, it is possible that \infty > (y* - N(x*))^2 > (y* - N(x'))^2 >  f(epsilon). The issue is that f(epsilon) can be less than infinty for a given epsilon.

The authors try to answer the questions on the theorems and definitions in the rebutall, and I have read the responses. I think some answers have resolved the concerns, but some questions still remain. In addition, I think there are other questions on the theorems and definitions to be answered. I think the authors need to provide a new definition and a new theorem statement so as to eliminate all possible concerns which are not all enumerated by reviewers.

   Overall, I think the paper needs much polishing before considering publication. While I do appreciate the proposed interpretation of adversarial examples, I can only recommend rejection for the current version.

**Justification For Why Not Higher Score:**

The theorem statements and definition are unclear and confusing. Not ready for publication.

**Justification For Why Not Lower Score:**

N/A

---

### Decision · Program_Chairs · 2024-01-16

Reject